# A Practical Procedure to Integrate the First 1:500 Urban Map of Valencia into a Tile-Based Geospatial Information System

Miriam Villar-Cano, María Jesús Jiménez-Martínez *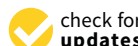 and Ángel Marqués-Mateu

Department of Cartographic Engineering, Geodesy and Photogrammetry, Universitat Politècnica de València, 46005 València, Spain

* Correspondence: mjjimenez@cgf.upv.es; Tel.: +34-6447-716-877

**Abstract:** The use of geographic data from early maps is a common approach to understanding urban geography as well as to study the evolution of cities over time. The specific goal of this paper is to provide a means for the integration of the first 1:500 urban map of the city of València (Spain) on a tile-based geospatial system. We developed a workflow consisting of three stages: the digitization of the original 421 map sheets, the transformation to the European Terrestrial Reference System of 1989 (ETRS89), and the conversion to a tile-based file format, where the second stage is clearly the most mathematically involved. The second stage actually consists of two steps, one transformation from the pixel reference system to the 1929 local reference system followed by a second transformation from the 1929 local to the ETRS89 system. The last stage comprises a map reprojection to adapt to tile-based geospatial standards. The paper describes a pilot study of one map sheet and results showed that the affine and bilinear transformations performed well in both transformations with average residuals under 6 and 3 cm respectively. The online viewer developed in this study shows that the derived tile-based map conforms to common standards and lines up well with other raster and vector datasets.

**Keywords:** coordinate transformation; Akaike information criterion; quality control; urban mapping; cartographic heritage; tile-based geospatial information systems

---

## 1. Introduction

Early maps are of invaluable importance for a wide range of applications such as landscape change analysis [1], territorial planning [2,3], urban development studies [3,4], and archaeological research [5]. Modern information technologies, particularly geographic information science and web geoservices, provide considerable potential in all of those geospatially-based studies [6,7]. In this paper we argue in favor of such geospatial technologies to improve the interpretation of digital versions of early maps with an example from the city of València in Spain.

At the end of the 1920s, València was a dispersed and unfinished city where different economic activities were shifted to the outskirts [3]. The urban planning in Valencia acted as a decisive factor for business location, affecting land value and other variables. For instance, railways had a clear influence on urban growth, but also in the configuration of the space itself, and the impact of urban transport was similar, especially in terms of the location of economic activity, urban mobility, and land revaluation [8].

In a municipal plenary session held on 2 July 1928, the València City council approved the production of the first accurate topographic (urban and rural) map to scale 1:500 by the Instituto Geográfico y Catastral (IGC), the former Spanish National Mapping Agency. This local initiative responded to the needs of policy makers who required better information about the development of the city, particularly in suburban areas.

The topographic survey was conducted according to the specific regulations of the IGC, dated 30 May 1928, with the exception of the map scale. Even though the scale value for urban maps in the Article 72 of the regulations was 1:2000, the final scale value used in this project was 1:500. The map had associated the corresponding notebooks of each cadastral polygon with the numbered list of plots and subplots, together with their areas, land uses, types of crops, and owners. According to Article 68, buildings, wells, water wheels, roads, ditches, paths, and other relevant topographic details had to be drawn in the map within each plot. The 1929 map was manually drawn in different colors: trees and gardens in green, water facilities in blue, contours in brown, properties in grey, stations of the second order network traverses in yellow, and house numbers in green.

Some years after completing the 421 map sheets of the project, the map was used as the essential geometric base for the development of the Urban Plan of València (1944–1946) [9]. The dates show that this project took a period of 15 years to complete, from 1929 to 1944, with an intervening civil war. The map covered about 174 km$^2$ and showed the true status of the territory, thus becoming an effective tool for the representation, analysis, comparison, and evolutionary studies of València and its immediate surroundings [3].

The technical quality and documentary value of the map was demonstrated in previous research after intensive fieldwork and recomputation of the underlying geodetic network. The observational and computational processes of the network, originally conducted almost one century ago, prove the high quality of the work [10]. The results were very encouraging and somehow justify our interest in improving the quality of the derived digital map by finding the best approach in terms of geometric transformations, with the goal of adapting the 1926 map to tile-based geospatial standards used in web platforms. In this line of research, the General Conference of UNESCO at its 32nd session [11] adopted the "Charter on the Preservation of Digital Heritage", thus recognizing the growing importance of digital heritage, its vulnerability, and the need for its preservation. One of the key conclusions of the conference was the need for promoting and widening access to information in the public domain through the organization, digitization, and preservation of heritage in digital format. Along the same lines, the International Cartographic Association (ICA) created in 2007 the Commission on Digital Technologies in Cartographic Heritage, whose aim is to encourage digital approaches to cartographic heritage [7].

In our study, we developed a workflow consisting of three stages. The first is the digitization of the 421 original map sheets. Digitization of paper historical maps is usually performed with high-resolution scanners, which produce large raster images. Four important parameters must be set in the digitization phase: resolution, pixel bit depth, color, and file format. A precise specification of these parameters for historical map digitization does not exist [6]. Resolution of 300 dpi and 24-bit RGB color depth has become a de facto standard for historical map digitization projects [6]. In our study, we used the Digibook Suprascan A0 cenital 10000 RGB with 3000 dpi scan resolution, leading to image files with a ground pixel size of 5 cm. The bit depth was 36-bit in the RGB system to capture all color information. Both TIFF master files and JPEG access files were produced. The 421 original map sheets were digitally scanned but only one map sheet was analyzed in order to describe the second and third stages of the proposed process.

The second stage is the most mathematically and computationally involved. The goal of this stage was to determine the best georeferencing procedure for the 1929 map by setting out a precise scheme to keep the quality of the original in the final digital map, which is known to be 10–15 cm [10]. This approach makes the future production of a complete digital map for public use easier. While the first and last stages are certainly common to other similar projects, the mid stage is highly specific to ours and takes advantage of the metric characteristics of the map, which are somewhat encompassed in the underlying triangulation network. The transformation of a map without losing its original metric properties is of particular importance, and it is for this reason that special attention must be paid to the preservation of geometric properties when modifying or transforming the map [12]. In [10], the authors computed the global transformation of the primary triangulation of the 1929 map to UTM-ETRS89 and

it was suggested that the rigorous geometric transformation of the 1929 map should undergo local transformations using ground truth data points. This is indeed the basis of the present study.

We tested a number of geometric transformations to find the best mapping function. Quality indicators have been proposed to choose the right transformation such as the least squares adjustment or the Akaike Information Criterion (AIC). It is well-known that least squares can be used as a quality control analytic tool in surveying engineering and digital mapping [13,14]. However, we believe that AIC fits better to our case study since it employs a comprehensive point of view that takes into account both the accuracy and the complexity of the transformation function, represented by the number of parameters [15]. In addition, the AIC allows the introduction of check points, i.e., points with known coordinates that are used for validation purposes only, independently from the model training computations.

The third and last stage of the process is the conversion to a tile-based file format. The development of web mapping technologies, network communications, and mobile positioning has greatly contributed to starting a new class of computer programs, which are key components for location-based services where digital maps play a key role [16,17]. In the particular case of web cartography, evolution has led to a common and accepted tile-based format, which is used, with minor changes, in all major web mapping services today [18].

The accuracy limit of the resulting digital urban map of València was calculated following the Positional Accuracy Standards for Digital Geospatial Data of the American Society for Photogrammetry and Remote Sensing (ASPRS). This standard is intended to be used by geospatial data providers and users to specify the positional accuracy requirements for final geospatial products.

The main purpose of this paper is to integrate an early urban map from 1929, originally printed on paper and made with high precision techniques of the time, into a modern geospatial database in digital format without loss of quality, and preserve that information for both interested technically-oriented users and future generations of scholars.

The paper is structured as follows. Section 2 provides the materials and methods for spatial procedures necessary to perform coordinate transformations of the horizontal datum. Section 2.3 deals with the procedure to create a tile-based map and its visualization on a web browser together with other geospatial layers. Section 3 evaluates the spatial accuracy obtained from the pixel to 1929 transformation and the 1929 to UTM-ETRS89 transformation, respectively. Section 3.3 is about using a tile map service. Finally, in Sections 4 and 5 the discussion and conclusions drawn, respectively, from the study are presented.

## 2. Materials and Methods

In the mid-1990s there was renewed interest among land surveyors, GIS experts, remote sensing researchers, and other geospatial practitioners in coordinate transformation methods. This was mainly due to the need for consolidating legacy geospatial datasets measured and processed in old coordinate systems with high accuracy [19].

One of the numerical procedures which are commonly required in the mapping sciences are the 2D linear transformations to convert from one Cartesian coordinate system into another [20]. A typical application of this type of 2D transformation is the overlay of digitized paper maps showing historic land use with modern geographic datasets within a GIS environment. Particular care has to be taken into account when digitizing old paper maps, especially with respect (but not restricted) to differential paper shrinking rates [21]. Proper material preservation, uniform coordinate grid coverage, and other factors help in reducing the effects of differential shrinking. These two specific conditions are met in the 1929 map sheets which have been always stored in the archives of the València city council and have a coordinate grid frame in every sheet defining a unique, continuous coordinate reference system.

The key point here is to find the best plane transformation for our case study which actually consists of two concatenated transformations. The first transformation converts from the pixel coordinate system of the scanned map (measured in terms of rows and columns of image pixel array) into the

geodetic coordinate system of the 1929 map. In a second step, the 1929 map coordinate values must be converted into a modern coordinate reference system, specifically the European Terrestrial Reference System of 1989 (ETRS89), the official system in Spain since 2012.

In this section we present the mathematical fundamentals of the most commonly used 2D transformations, which are the essential building blocks to define and determine which one fits best to transform between coordinate systems, thus ensuring the efficient transfer of information from one map to another. The selection of the proper transformation for a specific map requires a case-by-case assessment since it is impossible to develop a transformation that is optimal for all cases [19].

The assessment must take into account the parameter set resulting from every single transformation model against a specified acceptable quality level, which is a function of the accuracy of the map under study. A well-known fact is that the map scale is directly associated with an intrinsic error map. Before computer mapping and GIS technology existed, map sheets were drawn by hand, so that map scale itself was a significant contributor to the map accuracy due to practical aspects such as the width of the pen used to draw the graphics. In our case the map was drawn at a nominal scale of 1:500, therefore, a 0.2 mm pen generates 10 cm width lines at ground scale. Conversely, any object of size under 10 cm in ground units scales down to less than 0.2 mm on the map, and has no physical representation. Consequently, a value of 10 cm appears to be the acceptable error limit for the process of transforming the analog 1929 map into a digital georeferenced map.

Despite the fact that geometric quality in geospatial data is one of the most important factors for providers and users, and that the topic has been largely discussed in the literature from the research, standardization, and implementation standpoints, it is difficult to agree on a common or universal approach. For instance, the INSPIRE directive [22], which is the reference regulatory frame on spatial data infrastructures in the European Union, does not provide a priori data quality requirements, although it contains recommended preferences for data quality results. The International Organization for Standardization (ISO), an independent, non-governmental international organization, establishes the principles for describing the quality of geographic data, but it does not establish specific quality levels for digital maps (ISO 19157:2013) [23]. Along the same lines, the American Society for Photogrammetry and Remote Sensing (ASPRS) horizontal accuracy standards specifies the primary horizontal accuracy for digital data, including digital orthoimagery, digital planimetric data, and scaled planimetric maps [24]. This standard defines accuracy classes based on Root-Mean-Square-Error (RMSE) thresholds, in terms of their RMSE-X and RMSE-Y values. The recommendations for digital planimetric data produced from digital imagery, are based on current status of mapping technologies and best practices. The horizontal accuracy for digital planimetric data produced from digital imagery, and their equivalent map scales according to the legacy standards of the ASPRS 1990 and the United States National Map Accuracy Standards (NMAS) of 1947 are RMSE-X = 12.5 cm and RMSE-Y = 12.5 cm at the 68% confidence level, and RMSE-X = 30.6 and RMSE-Y = 30.6 cm at the 95% confidence level. Therefore, a suitable threshold value for an acceptable map quality level is 12.5 cm.

## 2.1. Two-Dimensional Coordinate Transformations

### 2.1.1. Similarity Transformation

The 2D similarity transformation has four parameters that define two translations, a rotation and a scaling factor between two reference systems preserving angles and distance ratios [25]. The assumption in the similarity transformation is that the scale factor is a single, unique value. This is a reasonable assumption to make in some cases, but it cannot be justified in others. For example, the scanning of the maps may be affected by deformation of the original paper sheets by stretching and shrinking, and it is not usually the same in all directions [15,19,21]. This mathematical model is too restrictive for the 1929 map, so we discarded it. We report on similarity transformation for the sake of completeness.

The formulas of the similarity transformation are [25]

$$X = a_0 + a_1 x + b_1 y,$$
$$Y = b_0 + b_1 x + a_1 y. \tag{1}$$

### 2.1.2. Affine Transformation

The affine model characterizes non-uniform deformations with different scale factors in the directions of the two coordinate axes as well as with the obliquity between the axes [24,25]. It is a six-parameter transformation based on two scale parameters (one for each coordinate axis direction), two translations and two rotations. The affine transformation allows independent scale and rotation adjustments in each coordinate axis, and is thus able to correct many effects from actual physical and practical causes. In scanned maps, it can correct for differential effects such as different paper shrinkage in each direction. In addition, this transformation can correct some errors caused by other side effects such as differences of datum and map projection between the source and target spaces of the map [15,21].

The formulas of the affine transformation are [25]

$$X = a_0 + a_1 x + a_2 y,$$
$$Y = b_0 + b_1 x + b_2 y. \tag{2}$$

The affine transformation requires more than three control points (i.e., points with known coordinates in both systems) to compute the transformation parameters with redundancy. A set of three points allows the exact determination of the parameters.

### 2.1.3. Bilinear Transformation

In the bilinear transformation two parameters are added, in addition to the six parameters of the affine transformation. Those new parameters can be understood as two angles between the coordinate axes. Transformations with six or eight parameters for two dimensions have become more common recently [25,26].

The bilinear transformation is similar to the affine transformation [25]:

$$X = a_0 + a_1 x + a_2 y + a_3 xy,$$
$$Y = b_0 + b_1 x + b_2 y + b_3 xy. \tag{3}$$

This transformation requires more than four control points in order to determine the eight coefficients with redundancy.

### 2.1.4. Polynomial Transformation

Second order (12-parameter) polynomials are often used for the correction of scanner data. In addition to first-order distortions, polynomials correct second-order distortions, like map projection [21]. This transformation method is also used in GIS environments to relate datasets that do not match with the similarity or affine transformations [19,21].

Non-linear deformation can be described by high degree polynomials. A polynomial of the second degree is given by [25]

$$X = a_0 + a_1 x + a_2 y + a_3 xy + a_4 x^2 + a_5 y^2,$$
$$Y = b_0 + b_1 x + b_2 y + b_3 xy + b_4 x^2 + b_5 y^2. \tag{4}$$

In general, the number of coefficients required to define a polynomial transformation of degree n is: $u = (n + 1) \cdot (n + 2)$. In order to determine the u coefficients, a minimum of u/2 control points are required. It should be noted that increasing the degree of the polynomial transformations leads to

smaller residuals, but can distort the output map and give unusable results if proper control point sets are not available which limits the use of polynomial transformations in some cases [26].

### 2.2. Least Squares Adjustment and Akaike Information Criterion

#### 2.2.1. Least Squares Adjustment

A set of transformation parameters can be properly determined by a least-squares solution, which minimizes the sum of the squares of the residuals at the control points. As a rule of thumb, the stronger the match between the control points coordinates, the lower the values of residuals. The models tested were the affine, polynomial, and bilinear transformation.

All models generate two independent equations per control point. The input data to a generic geometric transformation consists of several pairs of control points, with known coordinates in the source and target reference system, each pair representing some sort of geometric link between the two spaces.

The matrix form of the equation system and its solution is well known [14,27]:

$$A \cdot x = b + \upsilon, \tag{5}$$

where A is the coefficient matrix, x is the vector of unknowns, b is the vector of independent terms, and $\upsilon$ is the vector of residuals. The least squares method allows specific weighting for every observation equation. Since we did not find any suggestions for a specific weighting, we computed the adjustment with equally weighted observations as follows:

$$x = \left(A^T \cdot A\right)^{-1} \cdot A^T \cdot b. \tag{6}$$

The most interesting point of the least squares method with respect to the approximate methods used back in 1929 is the calculation of the variance-covariance matrix, which contains the precision information of the variables, that is, the precision of the transformation parameters. The expression of the variance-covariance matrix ($\Sigma_x$) is:

$$\Sigma_x = \sigma_o^2 \cdot \left(A^T \cdot A\right)^{-1}, \tag{7}$$

where $\sigma_o^2$ is the a posteriori variance of unit weight which is computed using the following formula:

$$\sigma_o^2 = \frac{\upsilon^T \cdot \upsilon}{n - u}, \tag{8}$$

where $\upsilon$ is again the vector of residuals, $n$ is the number of equations, and $u$ is the number of unknowns. In adjustment theory, the expression $n - u$ is usually referred to as the degrees of freedom of the system which equals the number of redundant equations in the model. The precision information of the variables and the residuals give us very valuable information for testing purposes.

#### 2.2.2. The Akaike Information Criterion

The Akaike Information Criterion (AIC) is a commonly used method for model comparison in statistical analysis. The criterion can be applied to any statistical model. Some authors have argued in favor of the AIC as a measure of the goodness of fit of the transformation model between different geodetic reference frames [15,26,28,29].

The AIC is an estimator of the relative quality of statistical models for a particular dataset. Given a collection of models for the dataset, the AIC estimates the quality of each model, relative to each of the other models, thus providing a means for model selection. In the present case the proposed models are the affine, bilinear, and second order polynomial models.

The residual sum of squares (RSS) from the least squares adjustment is a popular indicator for the suitability of the transformation method. Nevertheless, the RSS error may not be an ideal criterion because a transformation model with a large number of parameters will normally yield a smaller RSS error. However, a transformation model with a large number of parameters is highly sensitive to outliers and may incorrectly distort, stretch, or alter the system [15,26,30]. The AIC principle provides additional information for model comparison in a simple and convenient manner. We applied the AIC as a tool for selecting our best transformation model.

The AIC value for one or multiple fitted model objects can be obtained (assuming normally distributed errors, as in least squares estimation) with the following formula [31]:

$$\text{AIC} = 2{\cdot}\text{k} + \text{n}{\cdot}(\ln((2{\cdot}\text{pi}{\cdot}\text{RSS})/\text{n}) + 1), \tag{9}$$

where k is the number of estimated parameters in the model (for any least squares model with Gaussian residuals, the variance of the residuals distribution should be counted as one of the parameters), n is the number of observations, and RSS is the residual deviance as the sum of squared residuals.

Thus, AIC rewards goodness of fit (as assessed by the likelihood function), but also includes a penalty that is an increasing function of the number of estimated parameters. The penalty discourages overfitting, because increasing the number of parameters in the model usually improves the goodness of fit.

The practice of AIC requires a set of candidate models to compute their corresponding AIC values. There is always some information loss due to selecting a candidate model to represent the true model. We wish to select the model that minimizes that information loss from the candidate models. Thus, we cannot choose with certainty, but we can minimize the estimated information loss. AIC is useful for comparing models, but is not interpretable on its own.

The AIC may perform poorly in small datasets. Since in coordinate transformation problems we usually deal with small sample sizes, the second-order Akaike Information Criterion ($\text{AIC}_C$) should be used instead of the regular AIC as proposed by different authors [15,26,30,31]. The alternative version $\text{AIC}_C$ means AIC with a correction for small sets of observations.

The $\text{AIC}_C$ is defined as [31]

$$\text{AIC}_\text{c} = \text{AIC} + \left(\frac{2{\cdot}\text{k}{\cdot}(\text{k}+1)}{\text{n}-\text{k}-1}\right). \tag{10}$$

### 2.3. Creating a Tile-Based Geospatial System

The proper geometric transformation of the original paper sheets into digital image files is the previous step towards the creation of a tile-based geospatial system which feeds a tile map service (TMS). A TMS is by definition a network-based geospatial service that relies on tile-based formats [18]. The tile-based mapping approach differs from common raster-based image mapping in a number of ways that have been mentioned above (see Section 1), but the most important is the creation of different zoom levels which contain specific sets of rendered image files ready to be represented on the screen. Every single file is usually referred to as a tile.

In the example of this paper we used sheet 54 II, one of the map sheets of 1929, which was transformed into GeoTIFF format in the EPSG:25830 coordinate reference system. We conducted several tests to find the right transformation, as profusely described above in Sections 2.1 and 2.2.

The steps to create a tile set from a single image file consist of (1) reprojecting the original image into the EPSG:3857 reference system, and (2) creating the standard tile sets. Both tasks can be done using free and open source software systems. We used the QTiles plugin for QGIS which relies on OSGeo tools to conduct all image manipulations, although there are other alternatives to get the same results. The Qtiles plugin allows the definition of all relevant parameters in a single dialogue box (see Figure 1), the most important being the input data and the output tile format.

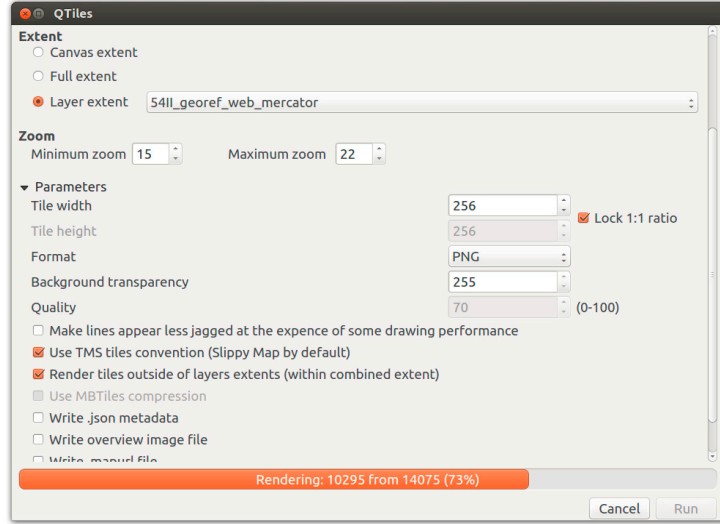

**Figure 1.** Graphical user interface of the QTiles plugin showing some relevant parameters.

Note that depending on the tile dimensions and the number of zoom levels the number of tiles varies. In this particular run, the plugin created more than 14,000 tiles which took about 2 min of computation time.

## 3. Results

### 3.1. Results of the Transformation from Pixel Reference System to 1929 Reference System

In this section the process to transform the pixel reference system to the 1929 local reference system is tested in a real case study. The emphasis is on the choice of the proper transformation method between the two coordinate systems.

#### 3.1.1. Data Points

The input data to the first geometric transformation consists of a set of coordinate pairs in the source (pixel) and target (1929) reference systems. In order to calculate the numeric coefficients seven points were selected (Table 1). In order to reduce the error of the pixel coordinates we only selected well defined intersection points on the coordinate grid frame (note that all control points have integer coordinate values corresponding to grid ticks, see also Figure 2). The internal coordinate grid was drawn by hand using pencil and the ticks are visible in some areas of the map as small crosses made with two thin lines. The grid frame was redrawn at a later stage using black ink which slightly increased the thickness of the frame ticks. The frame of all the 1929 map sheets, invariably falls directly on grid lines with integer coordinate values (Figure 2).

**Table 1.** Point coordinates in the source (pixel) and target (1929) reference systems.

| Point | X-Pixel | Y-Pixel | X-1929 (m) | Y-1929 (m) |
|-------|---------|---------|------------|------------|
| 1 | 989 | 990 | 24,950 | 34,950 |
| 2 | 3337 | 1000 | 25,150 | 34,950 |
| 3 | 987 | 1578 | 24,950 | 34,900 |
| 4 | 2749 | 1589 | 25,100 | 34,900 |
| 5 | 966 | 5708 | 24,950 | 34,550 |
| 6 | 2156 | 2765 | 25,050 | 34,800 |
| 7 | 2746 | 2178 | 25,100 | 34,850 |

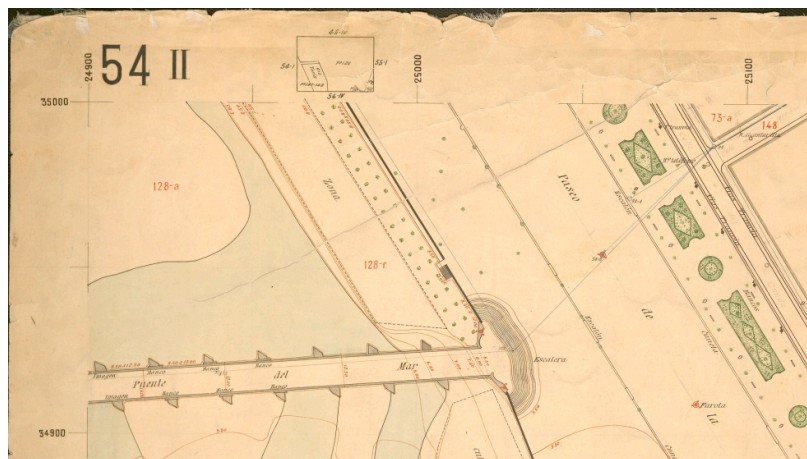

**Figure 2.** Grid frame corner of the map number 54 II. The X axis and the Y axis show the local coordinates, on this image from 24,900 to 25,100 m and from 35,000 to 34,900 m respectively.

### 3.1.2. Fitting Models

This section contains a summary of the quality figures of the study. Table 2 shows the number of parameters, the significant parameters together with the residual sum of squares (RSS), AIC and AICc values for each model transformation. According to Equations (9) and (10) the AICc value of the polynomial transformation ranges from −39.1814 to −50.0757. As the first equation term $2 \cdot k \cdot \left( \frac{n}{n-k-1} \right)$ is zero, the Akaike information criterion value is not valid for this model.

**Table 2.** Number of parameters, significant parameters, residuals sum of parameters (RSS) and values of AIC and AICc for each model.

| Model | Number of Parameters | Significant Parameters | RSS-X ($m^2$) | RSS-Y ($m^2$) | AIC X | AIC Y | AICc X | AICc Y |
|---|---|---|---|---|---|---|---|---|
| Affine | 6 | 6 | 0.028 | 0.029 | −12.785 | −12.539 | −11.029 | −10.783 |
| Bilinear | 8 | 7 | 0.008 | 0.023 | −17.121 | −12.162 | −19.798 | −12.406 |
| Polynomial | 12 | 5 | 0.0037 | 0.0008 | −18.938 | −29.832 | — | — |

The estimated parameter $p$-values, standard errors $\sigma$ from Equation (8), residuals, t values (t $= \frac{P}{\sigma}$), probability and significance of the affine and bilinear transformations are in Appendix A.

The choice of the most suitable model for georeferencing purposes was based on these numeric values. The bilinear transformation model has the minimum RSS, minimum AIC, minimum AICc, and residuals are all under the accuracy limit of 12.5 cm. Consequently, bilinear transformation appears to be the model that fits best to the dataset selected in this specific area, although one parameter has a not significant probability.

### 3.1.3. Checking the Selected Model

The values of the transformation parameters are derived by the least squares technique which provides statistical parameters (i.e., the variance of unit weight $\sigma^2$) to account for the internal precision rather than the external accuracy. Consequently, an independent set of control points was selected to check out the results of the transformation.

Using the bilinear transformation parameters and eight pairs of pixel and map coordinates we defined a set of eight check points and computed their transformed coordinates (Table 3, columns 4 and 5). We selected check points pertaining to grid coordinate tick set which all have known map coordinates (Table 3, columns 6 and 7). All differences between transformed coordinates and grid coordinates except one are lower than 10 cm, which proves that the selected model fits well.

**Table 3.** Transformed coordinates from pixel coordinates in bilinear model, grid coordinates, and differences in between.

| Point | X-Pixel | Y-Pixel | X-Transf. (m) | Y-Transf. (m) | X-Grid (m) | Y-Grid (m) | ΔX (m) | ΔY (m) |
|---|---|---|---|---|---|---|---|---|
| 1 | 1555 | 5711 | 24,999.983 | 34,550.080 | 25,000 | 34,550 | −0.017 | 0.080 |
| 2 | 1576 | 991 | 24,999.959 | 34,950.089 | 25,000 | 34,950 | −0.041 | 0.089 |
| 3 | 2159 | 2176 | 25,050.029 | 34,849.929 | 25,050 | 34,850 | 0.029 | −0.071 |
| 4 | 4510 | 1006 | 25,249.890 | 34,950. 008 | 25,250 | 34,950 | −0.110 | 0.008 |
| 5 | 4510 | 1596 | 25,249.997 | 34,900.078 | 25,250 | 34,900 | −0.003 | 0.078 |
| 6 | 3921 | 2184 | 25,199.982 | 34,850.048 | 25,200 | 34,850 | −0.018 | 0.048 |
| 7 | 5684 | 2194 | 25,350.020 | 34,849.998 | 25,350 | 34,850 | 0.020 | −0.002 |
| 8 | 2165 | 405 | 25,049.930 | 34,999.976 | 25,050 | 35,000 | −0.070 | −0.024 |

In order to compare the bilinear transformation with the affine transformation, we calculated the transformed coordinates using the affine parameters and obtained the differences (Table 4). The affine transformation fits worse than the bilinear transformation.

**Table 4.** Transformed coordinates from pixel coordinates in affine model, grid coordinates, and differences in between.

| Point | X-Pixel | Y-Pixel | X-Transf. (m) | Y-Transf. (m) | X-Grid (m) | Y-Grid (m) | ΔX (m) | ΔY(m) |
|---|---|---|---|---|---|---|---|---|
| 1 | 1555 | 5711 | 25,000.106 | 34,550.009 | 25,000 | 34,550 | 0.106 | 0.009 |
| 2 | 1576 | 991 | 24,999.939 | 34,950.099 | 25,000 | 34,950 | −0.061 | 0.099 |
| 3 | 2159 | 2176 | 25,050.070 | 34,849.904 | 25,050 | 34,850 | 0.070 | −0.096 |
| 4 | 4510 | 1006 | 25,249.763 | 34,950.008 | 25,250 | 34,950 | −0.236 | 0.008 |
| 5 | 4510 | 1596 | 25,250.008 | 34,900.071 | 25,250 | 34,900 | 0.008 | 0.071 |
| 6 | 3921 | 2184 | 25,200.101 | 34,849.978 | 25,200 | 34,850 | 0.101 | −0.022 |
| 7 | 5684 | 2194 | 25,350.217 | 34,849.884 | 25,350 | 34,850 | 0.217 | −0.116 |
| 8 | 2165 | 405 | 25,049.847 | 35,000.022 | 25,050 | 35,000 | −0.153 | 0.022 |

## 3.2. Results of the Transformation from 1929 Reference System to UTM-ETRS89 Reference System

In this section the process to transform the 1929 local reference system reference system to UTM-ETRS89 reference system is tested in another real case study to choose the proper transformation method between both coordinates systems.

### 3.2.1. Data Points

In order to determine the coefficients of the transformation we used seven points with known coordinates in both the 1929 reference system and the UTM-ETRS89 reference system (Table 5).

**Table 5.** Point coordinates in the source (1929) and target (UTM-ETRS89) reference systems.

| Point | X-1929 (m) | Y-1929 (m) | X-UTM-ETRS89 (m) | Y-UTM-ETRS89 (m) |
|---|---|---|---|---|
| Mislata | 20,310.3 | 35,452.4 | 722,137.615 | 4,372,684.928 |
| Sancho | 27,377.67 | 31,860.15 | 729,305.691 | 4,369,296.395 |
| Miguelete II | 23,915.43 | 35,480.6 | 725,740.934 | 4,372,816.608 |
| Pechina | 22,514.06 | 35,662.39 | 724,334.668 | 4,372,958.164 |
| 298 | 24,453.03 | 35,745.74 | 726,270.642 | 4,373,097.113 |
| Puente del Mar | 25,029.35 | 34,925.5 | 726,870.327 | 4,372,293.579 |
| Puente del Mar II | 24,849.59 | 34,912.4 | 726,690.931 | 4,372,275.460 |

The coordinate values used in this analysis come from documents and field books of the 1929 project files (source reference system) and field measurements collected with global navigation satellite system (GNSS) equipment (target reference system). It should be noted the high difficulty to collect

reliable data for this transformation, which requires locating a set of point marks made 90 years ago. We conducted most of the necessary fieldwork in previous research [10] and it turned out that all the collected points were valid for the present study. Figure 3 shows one of those control point marks (coded vertex 298 in the project files), both in the field and on the map.

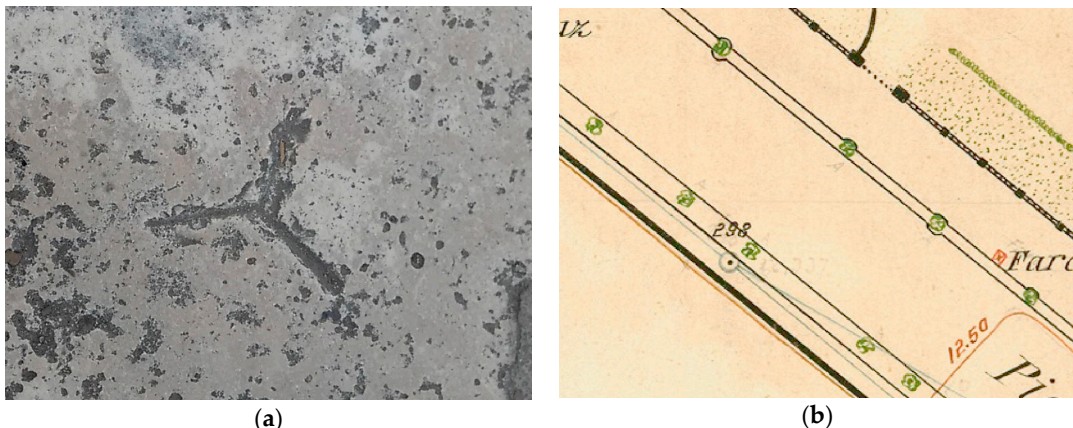

(**a**)　　　　　　　　　　　　　　　　　　　　　　　　　　　　　(**b**)

**Figure 3.** Two representations of vertex 298 (**a**) the original mark carved on the pavement; (**b**) location of the vertex 298 on the 1929 map.

### 3.2.2. Fitting Models

This section includes the number of parameters, the significant parameters, the residual sum of squares (RSS) values, AIC and AICc values for each model transformation (Table 6). According to Equations (9)–(10), the AICc is not a representative value for the polynomial model, as discussed in Section 3.1.2.

**Table 6.** Number of parameters, significant parameters, residuals sum of parameters (RSS), and values of AIC and AICc for each model.

| Model | Number of Parameters | Significant Parameters | RSS-X ($m^2$) | RSS-Y ($m^2$) | AIC X | AIC Y | AICc X | AICc Y |
|---|---|---|---|---|---|---|---|---|
| Affine | 6 | 6 | 0.014 | 0.012 | −17.637 | −18.716 | −15.88 | −16.960 |
| Bilinear | 8 | 6 | 0.014 | 0.011 | −15.637 | −17.325 | −1.881 | −3.6329 |
| Polynomial | 12 | 4 | 0.239 | 1.41 | 10.231 | 22.685 | —— | —— |

The *p*-values, standard error $\sigma$ according Equation (8), residuals, t value (t $= \frac{P}{\sigma}$), probability and significance of the affine, bilinear, and polynomial transformations can be found in Appendix B.

In the 1929 to UTM-ETRS89 transformation, the affine model has the minimum RSS. All parameters have significant probabilities, minimum AIC, minimum AICc, and residuals below the accuracy limit of 12.5 cm. Consequently, this model appears to be the best transformation model for the dataset selected. However, the bilinear transformation has very similar results, except for the AIC value, which is slightly lower, and the AICc which is also slightly lower than its affine counterpart (cfr. Table 6).

### 3.2.3. Checking the Selected Model

Similarly to the control point set, the collection of the check point set requires fieldwork to conduct the geodetic survey on a number of selected points. We selected four vertices coded 67A, 86-A, 299, and 299A in the original survey project (see Table 7) which were part of the second order transverse network. These vertices indeed meet the requirements to become a rigorous validation vertex. Their coordinates are known in both the origin (1926) and target (UTM-ETRS89) spaces using two different and separate procedures. The origin coordinates were retrieved from the project files, whereas the ETRS89 coordinates were obtained using GNSS equipment (see Figures 4 and 5), that is, an independent source of higher accuracy, as recommended in the ASPRS horizontal accuracy standards [23]. Using the affine and bilinear parameter sets we obtained the UTM-ETRS89 coordinates of the check vertices 67A, 86-A, 299, and 299A.

**Table 7.** Vertices from the field books used to check the mathematical model.

| Point | X-1929 (m) | Y-1929 (m) |
|-------|------------|------------|
| 67A | 24,736.67 | 35,580.69 |
| 86A | 25,008.74 | 35,060.06 |
| 299 | 24,542.65 | 35,672.22 |
| 299A | 24,530.53 | 35,701.03 |

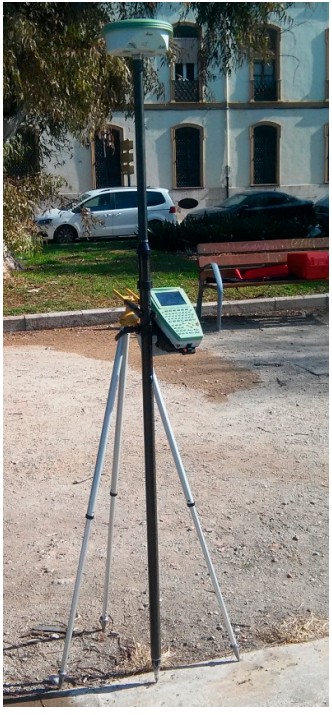

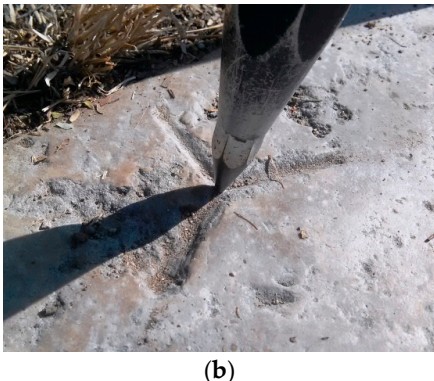

(b)

(a)

**Figure 4.** (**a**) The global navigation satellite system (GNSS) antenna and receptor during the resurvey of the vertex 86A; (**b**) the original mark of the vertex 86A.

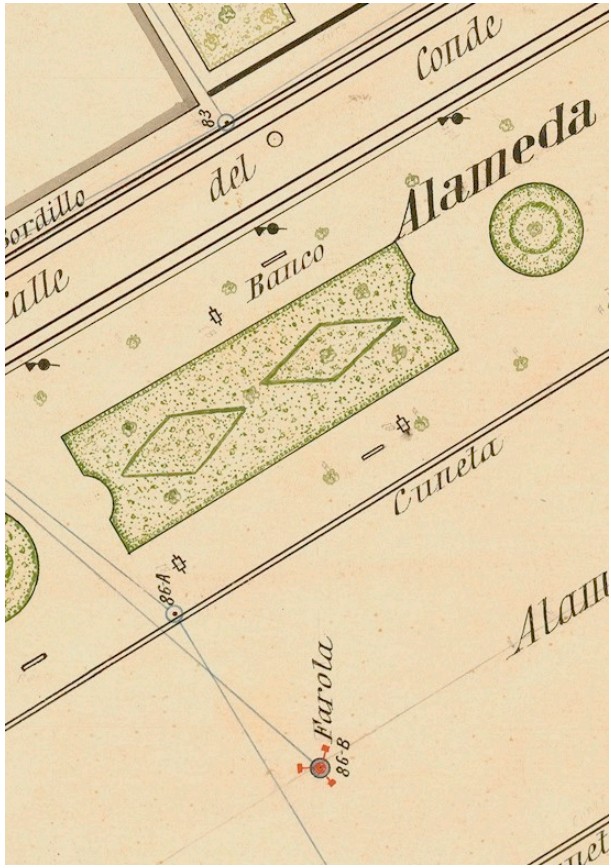

**Figure 5.** Sample of the 1929 map sheet with the location of the vertex 86A.

Table 8 shows the coordinate differences between the 1929 and the GNSS coordinates of all check points for the affine and bilinear models. Note that those differences are very similar in both mathematical models.

**Table 8.** Differences between UTM-GNSS coordinate and UTM-transformed coordinates.

| Model | Vertex | X-UTM GNSS (m) | Y-UTM GNSS (m) | X-UTM Transf. (m) | Y-UTM Transf. (m) | ΔX(m) | ΔY(m) |
|---|---|---|---|---|---|---|---|
| Affine | 67A | 726,510.529 | 4,372,928.407 | 726,510.601 | 4,372,928.482 | −0.070 | −0.069 |
| Bilinear | 67A | 726,510.529 | 4,372,928.407 | 726,510.602 | 4,372,928.476 | −0.074 | −0.060 |
| Affine | 86A | 726,845.851 | 4,372,427.521 | 726,845.854 | 4,372,427.567 | −0.003 | −0.046 |
| Bilinear | 86A | 726,845.851 | 4,372,427.521 | 726,845.838 | 4,372,427.576 | −0.003 | −0.055 |
| Affine | 299 | 726,362.333 | 4,373,026.171 | 726,362.378 | 4,373,026.206 | −0.045 | −0.035 |
| Bilinear | 299 | 726,362.333 | 4,373,026.175 | 726,362.378 | 4,373,026.196 | −0.045 | −0.026 |
| Affine | 299A | 726,349.437 | 4,373,054.591 | 726,349.465 | 4,373,054.661 | −0.028 | −0.070 |
| Bilinear | 299A | 726,349.437 | 4,373,054.591 | 726,349.471 | 4,373,054.650 | −0.034 | −0.059 |

### 3.3. Using a Tile Map Service (TMS)

The tile set, i.e., the tile-based geoinformation system properly, is at the core of a TMS. In our project, we chose the standard TMS disk scheme consisting of a single 256 × 256 PNG per tile, where the tiles are stored in a specific folder structure (there are other options based on more complex disk formats). The concept of the TMS file hierarchy is illustrated in Figure 6 which contains details of the former Aragón railway station in zoom levels 17, 18, 19, and 20. Every zoom level splits the whole map differently according to a systematic tile structure. The rationale behind the tile structure across zoom levels is that a given tile at zoom n generates four new tiles at zoom n + 1. For instance, if the first tile (left tile, zoom 17) in Figure 6 is split using a 2 × 2 grid, four new 128 × 128 tiles are created.

One of those four new tiles (the upper right tile) is the second tile in Figure 6, which is scaled up to 256 × 256 pixel dimensions, with the resulting increase in the details of the map elements. This same procedure is used to create the third and fourth tiles in Figure 6.

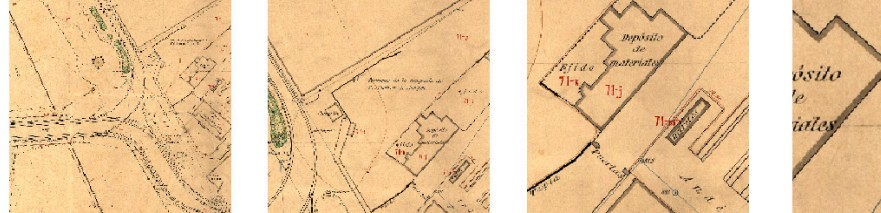

**Figure 6.** A detail of the former Aragón Railway Station at zoom levels 17 (left) to 20 (right). Note the increasing detail of the map elements in upper levels.

In the TMS approach, the client application is responsible for collecting the required tile subset based on the current position of the user and the zoom level, which is also set by the user. Map viewers retrieve all those tiles to fill up the screen with map content in the background, and need no direct user interaction, with the exception of setting the zoom level and location.

The usefulness of historic TMSs for historians, urban geographers, and engineers stems from their ability to mix with current data for different purposes. For visual, approximate comparisons, the overlap of the 1929 map with other existing TMS servers is suffice. Figure 7 contains a screenshot of a simple Leaflet viewer that provides such an overlap with some degree of transparency which improves the interpretation. For rigorous studies, the overlap of engineering-grade vector data and some additional editing tools are necessary. Typical use cases include setting out missing urban structures such as roads, buildings, and other urban facilities; or defining the boundary of urban plots in legal matters. Interested users can visit http://personales.upv.es/amarques/TMS/ijgi.html to experiment with the 54II map sheet, OpenStreetMap data, and a vector layer of the study area.

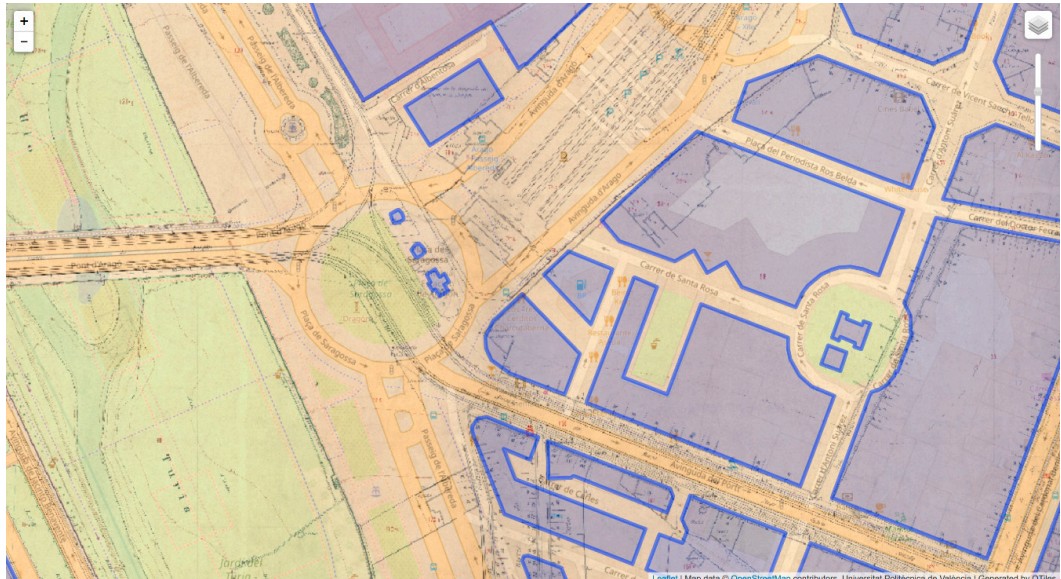

**Figure 7.** Screenshot of the viewer combining the 1929 map together with OpenStreetMap and vector data. The viewer allows replacing OpenStreetMap layer with an orthoimage.

## 4. Discussion

The main purpose of this paper is to integrate the first 1:500 urban map of València into modern geospatial databases in digital format without loss of quality. We designed a three-stage workflow

that worked well to achieve our goal. The first stage (scanning of paper maps) was done in previous research, so that the present paper focuses on stage 2 (geometric transformation from scanned map sheets to a modern reference system) and stage 3 (conversion to a tile-based image format).

In the process of producing the digital georeferenced map, the control and check datasets and the error of every stage must be rigorously supervised in order to preserve the original information for prospective users. We used the proper and most suitable transformations to ensure correct conversion between the different coordinate systems and correct error estimation of the transformation parameters. As previously described, the second stage consists of two successive transformations, the first to convert from the pixel reference system to the 1929 local reference system, and the second transformation to convert from the 1929 local to the UTM-ETRS89 system, the official system in Spain since 2012.

The choice of how many parameters should be used in the map transformations is vital. In order to choose the proper transformation, a refined Akaike Information Criterion (AICc) was used together with the least squares processing.

The production of the urban map of València spanned from 1929 to 1944 and resulted in 421 map sheets covering about 174 km$^2$. This large area suggests the use of local transformations in the 1929 to UTM-ETRS89 stage rather than a global transformation to reduce transformation residuals. While this approach sounds interesting to explore, our test on sheet 54II using a global transformation gave satisfactory results in terms of accuracy and image quality

We examined different types of 2D transformations based on a combination of parameters that have to be computed from selected control datasets. Results for the first and second transformations showed that bilinear and affine models respectively have appeared as the best local transformations to bring the 1929 data to modern coordinate reference systems. Actually, the choice between the affine and bilinear models is somewhat arbitrary since they yield very similar results, which ultimately proves the high quality of the control and check data.

In order to verify the accuracy, we also selected independent points to check out the results of the local transformations. All differences between transformed coordinates and their corresponding check points were lower than 11 cm, which meets the ASPRS threshold error (12.5 cm) for the quality of digital planimetric data derived from digital imagery. Again, this value confirms that the selected models fit very well to real datasets and the digital versions of the 1929 map can be used by the city council services in routine production.

The creation of the tile-based map in stage 3 was straightforward provided the quality of the transformed data from stage 2. Existing open and free software tools suited well to our needs that was to create a tile set of the 1929 map in the standard tile format. This tile-based geospatial system is the basis of a tile map service (TMS) with full map editing capabilities. In our opinion, the feasibility of such web service has great potential in several departments of the València city council and it would be worth exploring this issue together with specialized staff in future projects.

## 5. Conclusions

In this paper we sought to create a tile-based geospatial system from the first urban map of València (1929–1944) with emphasis on geometric quality. The results presented in this paper show that it is indeed possible to create such information system with a precise workflow and numeric tools to implement suitable quality control tests.

Well-known statistical tools such as the least squares adjustment method and the Akaike Information Criterion allowed rigorous quality control on the relevant stages of the process. The result was a digital copy of the original paper map in the ETRS89 coordinate reference system which conforms to the currently official system in our country, Spain. The resulting digital version of the map also allowed the creation of a tile-based dataset.

We wrote a simple viewer to experiment with one 1929 map sheet in combination with other online tile-based and vector datasets. Although this viewer is just a proof of concept, it shows the feasibility and potential of a complete system which can be built onto this small tool.

The results of the present exploratory work allows us to establish a method to publish in the future a completely operating mapping system for public use which will include the 421 map sheets.

In summary, this paper shows that early urban maps are still valid today to solve practical problems that would be unsolvable otherwise and can be useful in the context of historical geospatial information systems tile map services with Tms.

**Author Contributions:** Conceptualization, Miriam Villar-Cano, María Jesús Jiménez-Martínez; Methodology, María Jesús Jiménez-Martínez, Ángel Marqués-Mateu; Software, Ángel Marqués-Mateu; Validation, Miriam Villar-Cano, María Jesús Jiménez-Martínez; Formal Analysis, María Jesús Jiménez-Martínez; Investigation, Miriam Villar-Cano, María Jesús Jiménez-Martínez, Ángel Marqués-Mateu; Resources, María Jesús Jiménez-Martínez; Data Curation, María Jesús Jiménez-Martínez; Writing—Original Miriam Villar-Cano, María Jesús Jiménez-Martínez; Writing—Review and Editing, María Jesús Jiménez-Martínez, Ángel Marqués-Mateu; Visualization, Ángel Marqués-Mateu; Supervision, María Jesús Jiménez-Martínez, Ángel Marqués-Mateu.

**Funding:** This research received no external funding.

**Acknowledgments:** We would like to thank former Manuel Chueca, founder of the School of Geodesy, Cartography, and Surveying Engineering at the Universitat Politècnica de València in Spain, for providing bibliographic materials and a unique insight into the map of 1929. We also thank the staff at the València City Council for their kindness, help, and support, especially Cristina Sigalat and Juan Sáiz. Finally, the authors are sincerely thankful to the editor and two anonymous reviewers for their detailed comments and suggestions that helped the authors to considerably improve the manuscript into its final form as well as its readability.

**Conflicts of Interest:** The authors declare no conflict of interest.

## Appendix A

This appendix is a section that contains details and supplementary data about the affine and bilinear pixel to 1929 transformation: the estimate parameter values p, standard error $\sigma$ according Equation (8), t value ($t = \frac{P}{\sigma}$), probability and significance codes are listed in Tables A1 and A2 and residuals R in Tables A3 and A4.

**Table A1.** Affine transformation parameter value p, standard error $\sigma$, t value ($t = \frac{P}{\sigma}$), probability and significance codes.

| Parameter | Parameter Value p | $\sigma$ | t | Pr(>|t|) | Signific Code [1] |
|---|---|---|---|---|---|
| $a_0$ | 24,865.338377459 | $1.086 \times 10^{-1}$ | $2.28 \times 10^5$ | $<2 \times 10^{-16}$ | *** |
| $a_1$ | 0.0851460419891 | $3.689 \times 10^{-5}$ | 2308.16 | $2.11 \times 10^{-13}$ | *** |
| $a_2$ | 0.0004141569379 | $2.249 \times 10^{-5}$ | 18.41 | $5.12 \times 10^{-5}$ | *** |
| $b_0$ | 35,033.426316412 | $1.108 \times 10^{-1}$ | $3.161 \times 10^5$ | $<2 \times 10^{-16}$ | *** |
| $b_1$ | 0.0004270659313 | $3.763 \times 10^{-5}$ | 11.3 | $3.44 \times 10^{-4}$ | *** |
| $b_2$ | $-0.08476298350885$ | $2.294 \times 10^{-5}$ | $-3.69 \times 10^3$ | $3.22 \times 10^{-14}$ | *** |

[1] The significance codes of the linear model parameter are 0 = ***; 0.001 = **; 0.01 = *.

**Table A2.** Bilinear transformation parameter value p standard error $\sigma$, t value ($t = \frac{P}{\sigma}$), probability and significance codes.

| Parameter | Parameter Value p | $\sigma$ | t | Pr(>|t|) | Signific Code [1] |
|---|---|---|---|---|---|
| $a_0$ | 24,865.233705395 | $7.958 \times 10^{-2}$ | 312,461.330 | $<2 \times 10^{-16}$ | *** |
| $a_1$ | 0.0852476013224 | $4.512 \times 10^{-5}$ | 1889.454 | $3.27 \times 10^{-10}$ | *** |
| $a_2$ | 0.0004870540513 | $3.115 \times 10^{-5}$ | 15.637 | 0.000568 | *** |
| $a_3$ | $-0.00000006622813$ | $2.515 \times 10^{-8}$ | $-2.633$ | 0.078114 | .. |
| $b_0$ | 35,033.486118114 | $1.305 \times 10^{-1}$ | 268,360.940 | $<2 \times 10^{-16}$ | *** |
| $b_1$ | 0.0003690426076 | $7.401 \times 10^{-5}$ | 4.986 | 0.0155 | * |
| $b_2$ | $-0.08480463140732$ | $5.110 \times 10^{-5}$ | $-1.659 \times 10^3$ | $4.82 \times 10^{-10}$ | *** |
| $b_3$ | 0.0000000378377 | $4.126 \times 10^{-8}$ | 0.917 | 0.4267 | . |

[1] The significance codes of the linear model parameter are 0 = ***; 0.01 = *; 0.05 = ..; 0.1 = .

**Table A3.** Residual values of affine transformation.

| Residuals | $R_x$ (m) | $R_y$ (m) |
|:---:|:---:|:---:|
| $R_1$ | 0.042 | 0.067 |
| $R_2$ | 0.115 | −0.088 |
| $R_3$ | −0.031 | −0.092 |
| $R_4$ | −0.063 | 0.088 |
| $R_5$ | 0.047 | −0.012 |
| $R_6$ | −0.058 | 0.023 |
| $R_7$ | −0.051 | 0.015 |

**Table A4.** Residual values of bilinear transformation.

| Residuals | $R_x$ (m) | $R_y$ (m) |
|:---:|:---:|:---:|
| $R_1$ | 0.031 | 0.068 |
| $R_2$ | 0.029 | −0.039 |
| $R_3$ | −0.039 | −0.088 |
| $R_4$ | −0.064 | 0.089 |
| $R_5$ | 0.002 | 0.014 |
| $R_6$ | 0.021 | −0.023 |
| $R_7$ | 0.012 | −0.021 |

## Appendix B

This appendix is a section that contains details and supplementary data about the affine and bilinear 1929 to UTM-ETRS89 transformation: the estimate parameter values p, standard error $\sigma$ according Equation (8), t value ($t = \frac{P}{\sigma}$), probability and significance codes are listed in Tables A5 and A6 and residuals R in Tables A7 and A8.

Using the affine and bilinear mathematical model to resolve the least squared adjustment this Matlab warning appears: Matrix is close to singular. Results may be inaccurate. It would be preferable to avoid the singular matrix A, so the mathematical models (1) and (2) were replaced by:

$$
\begin{aligned}
X &= a_0 \cdot 725907.258286 + a_1 x + a_2 y, \\
Y &= b_0 \cdot 4372203.178142 + b_1 x + b_2 y,
\end{aligned}
\tag{11}
$$

$$
\begin{aligned}
X &= a_0 \cdot 725907.258286 + a_1 x + a_2 y + a_3 xy, \\
Y &= b_0 \cdot 4372203.178142 + b_1 x + b_2 y + b_3 xy.
\end{aligned}
\tag{12}
$$

Constant values 725,907.258286 and 4,372,203.178143 represent two translations of the coordinate axes.

**Table A5.** Affine transformation parameter value p, standard error $\sigma$, t value ($t = \frac{P}{\sigma}$), probability and significance codes.

| Parameter | Parameter Value p | $\sigma$ | t | Pr(>\|t\|) | Signific Code [1] |
|:---:|:---:|:---:|:---:|:---:|:---:|
| $a_0$ | 0.9682348838144 | $9.585 \times 10^{-6}$ | $505.01 \times 10^3$ | 0 | *** |
| $a_1$ | 0.9996937418504 | $5.521 \times 10^{-5}$ | $54.32 \times 10^3$ | 0 | *** |
| $a_2$ | −0.02863532893969 | $8.926 \times 10^{-5}$ | −961.229 | $7.03 \times 10^{-12}$ | *** |
| $b_0$ | 0.9918701185392 | $1.441 \times 10^{-6}$ | $3.44 \times 10^6$ | 0 | *** |
| $b_1$ | 0.0287069419929 | $4.998 \times 10^{-5}$ | $1.72 \times 10^3$ | $6.80 \times 10^{-13}$ | *** |
| $b_2$ | 0.9997689179512 | $8.080 \times 10^{-5}$ | $37.12 \times 10^3$ | 0 | *** |

[1] The significance codes of the linear model parameter are 0 = ***; 0.001 = **; 0.01 = *.

**Table A6.** Bilinear transformation parameter value p, standard error $\sigma$, t value (t $= \frac{P}{\sigma}$), probability and significance codes.

| Parameter | Parameter Value p | $\sigma$ | t | Pr(>\|t\|) | Signific Code [1] |
|---|---|---|---|---|---|
| $a_0$ | 0.9682339291628 | $4.259 \times 10^{-5}$ | $2.273 \times 10^4$ | $1.87 \times 10^{-13}$ | *** |
| $a_1$ | 0.99971906885450 | 0.0011283 | $8.860 \times 10^2$ | $3.17 \times 10^{-9}$ | *** |
| $a_2$ | $-0.02861583497542$ | $8.690 \times 10^{-4}$ | $-32.926$ | $6.15 \times 10^{-5}$ | *** |
| $a_3$ | $-0.00000000071164$ | $3.170 \times 10^{-8}$ | 0.4314 | 0.6953 | . |
| $b_0$ | 0.99187279797093 | $6.217 \times 10^{-6}$ | $1.595 \times 10^5$ | $4.441 \times 10^{-16}$ | *** |
| $b_1$ | 0.02827912024804 | $9.919 \times 10^{-4}$ | 28.5093 | $9.475 \times 10^{-5}$ | *** |
| $b_2$ | 0.99943949643420 | $7.640 \times 10^{-4}$ | $1.308 \times 10^3$ | $9.853 \times 10^{-10}$ | *** |
| $b_3$ | 0.00000001202562 | $2.787 \times 10^{-8}$ | 0.4314 | 0.695 | . |

[1] The significance codes of the linear model parameter are 0 = ***; 0.001 = **; 0.01 = *; 0.1 = .

**Table A7.** Residual values of affine transformation.

| Residuals | $R_x$ (m) | $R_y$ (m) |
|---|---|---|
| $R_1$ | 0.004 | 0.011 |
| $R_2$ | $-0.002$ | 0.007 |
| $R_3$ | $-0.097$ | 0.017 |
| $R_4$ | 0.022 | $-0.021$ |
| $R_5$ | 0.038 | 0.023 |
| $R_6$ | $-0.016$ | 0.051 |
| $R_7$ | 0.050 | $-0.087$ |

**Table A8.** Residual values of bilinear transformation.

| Residuals | $R_x$ (m) | $R_y$ (m) |
|---|---|---|
| $R_1$ | $-0.003$ | $-0.015$ |
| $R_2$ | 0.001 | $-0.001$ |
| $R_3$ | 0.097 | $-0.014$ |
| $R_4$ | 0.023 | 0.032 |
| $R_5$ | $-0.038$ | $-0.011$ |
| $R_6$ | 0.016 | $-0.063$ |
| $R_7$ | $-0.049$ | 0.073 |

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
