# Peer review of "A Practical Procedure to Integrate the First 1:500 Urban Map of Valencia into a Tile-Based Geospatial Information System"

_ijgi, doi:10.3390/ijgi8090378_

Round 1
Reviewer 1 Report
Scientific soundness of the paper is very low, as well as originality (novelty) of most of the used methods. I consider the practical part, the chosen method and the form of some parts of the text to be very problematic.
I am fascinated by the fact that authors are trying to create "tile-based geospatial system" containing only ONE sheet of the original map. The application is available online, but what an application can bring to users (which should be historians, urban geographers, and engineers) if the application allows only switching between 1 sheet of old map, footprints of few buildings, OSM and orthophoto map.
Described map (1929) was constructed without a mathematical basis?!? They are not mentioned in the paper. If so, it is not a map but a sketch. In general, the knowledge of mathematical cartography (projections etc.) from the 1920s is still used today (at least in some other European countries).
I'm not sure if the procedure described in this article will be applicable to all 421 map sheets. If I understand the procedure described well, it would be necessary to collect at least 421 x 8 ground control points.
Some parts of the text are full of very general information that is not entirely related to the goal of the paper. These parts of the text do not fit into the scientific text. For example:
· 6th paragraph of Instruction (rows 59 – 62)
· Second part of 11th paragraph of Introduction (rows 109 – 120)
· 1st paragraph in section Materials and methods (rows 138 – 141)
· 2.1.1, 2.1.2, 2.1.3 and 2.1.4 (rows 196 – 240) - Information in these paragraphs is at the level of the first year of geoinformatics studies
· 1st paragraph in section 3.3 Using a tile map service
Other comments:
· Title: “tile-based geospatial system” is very tricky term. Did you mean "raster GIS" or "tile-based GIS"?
· Title: This paper is not about "integration of the first 1:500 urban map", but about "integration of one map sheet"
· row 53: " ... 421 sheets of the project ..."; rows 478 - 479: " ... in 421 maps ..." Uniform terminology should be used. I think the correct term is a "map sheet(s)".
· Material and methods: INSPIRE does not contains data quality rules, because it is out of the scope of this directive. On the contrary, national mapping agencies of different countries (also from Europe) use such standards.
· 3.2.1 Data points - Is this chapter about ground control points? The correct terms should be used.
· Tables and their titles should be formatted better - Table 1 and Table 5.
· What is the source of the vector layer of building footprints in Fig. 7?
· I personally disagree with the last sentence in the penultimate paragraph. The created application is a proof-of-concept, but the real potential or feasibility of this system is for me very low.
· TMS (e.g. row 498) or Tms (row 516)?
Author Response
Dear Reviewer 1:
Thanks very much for the careful readings of the first version of the manuscript. Please, find below our replies to this review.
I am fascinated by the fact that authors are trying to create "tile-based geospatial system" containing only ONE sheet of the original map. The application is available online, but what an application can bring to users (which should be historians, urban geographers, and engineers) if the application allows only switching between 1 sheet of old map, footprints of few buildings, OSM and orthophoto map.
The goal of our study was to set a procedure or methodology to convert early surveying maps printed on paper to a modern digital format. Probably, this point was not clearly explained in the first version of the manuscript. We seek to work it out in these replies.
The fact that we present one only sheet is not relevant for the purpose of the study. The difference between processing one sheet or multiple sheets is a ‘production’ issue and not a ‘research’ issue. Therefore, the fundamental finding of the paper is to find the best approach in terms of geometric transformations, not to produce a complete map for public use. We were very accurate to test several geometric transformations and select the one that worked the best.
The results of that exploratory work allowed us to establish a method to keep the quality of the original in the final digital map, which is known to be 10-15 cm according to previous research conducted by the same authors (reference [10]).
Currently, we are in conversations with the Ajuntament de València (the City Council Authorities) on behalf of our department at UPV. The objective of those conversations is to sign an agreement to provide means (staff and funding) to conduct all that production work and publish a completely operating mapping system. The link provided in the manuscript was just a quick resource for reviewers to get the grasp of the system.
We added new contents in lines 66-69, 88-90 and 96-99 of the Introduction section to improve the explanation and understanding.
Described map (1929) was constructed without a mathematical basis?!? They are not mentioned in the paper. If so, it is not a map but a sketch. In general, the knowledge of mathematical cartography (projections etc.) from the 1920s is still used today (at least in some other European countries).
In a previous published paper by the same authors (referenced as [10] in the manuscript) we presented the geodetic survey work that was carried out for the primary triangulation network of the 1:500 urban map. There was indeed a big deal of mathematical background in the original survey which is rigorously documented in [10]. We also recomputed the work of 1929 using computer programs that mimic the computations done by hand in 1929.
In [10] we report on (1) Coordinate reference system: we assumed that the geographical coordinates were in the Madrid datum and the Tissot projection, still used in 1929, (2) Fourth order Primary triangulation process: baseline length and baseline orientation, triangulation, global transformation of the primary triangulation to UTM-ETRS89.
In our manuscript (lines 60-64) we reference this paper and emphasise the high quality of the map under study. We did not want to extend the manuscript with that information since it is already available to the readers. Indeed, the manuscript under review was derived as a “future research” line we found when working on paper [10]. The reviewers can get the paper from the editor if necessary, we have provided it him.
I'm not sure if the procedure described in this article will be applicable to all 421 map sheets. If I understand the procedure described well, it would be necessary to collect at least 421 x 8 ground control points.
The reviewer is right. This procedure is applicable to all 421 sheets.
However, the number of control points per sheet is variable. Affine and bilinear transformations require a minimum of 4-5 control points to get a mean squared solution. In the particular case of our case study, the 4 control points extend over 4 sheets, so that the same transformation parameters can applied to those 4 sheets.
The first transformation (pixel-1929) relies on the coordinate grid available in the original maps. The second transformation (1929-UTM/ETRS89) is based on ground points from triangulations, traverses, and other surveying works that must be located in the field works. This can be a practical limitation since some points have been lost. In that event, a convenience solution would be to use the transformation parameters of the adjoining sheets, instead of a unique, global transformation which was the first approach suggested in [10].
Some parts of the text are full of very general information that is not entirely related to the goal of the paper. These parts of the text do not fit into the scientific text. For example:
6th paragraph of Instruction (rows 60 – 63)
This paragraph was removed.
Second part of 11th paragraph of Introduction (rows 114 – 129)
This paragraph was edited and most of its contents removed. We left the three bibliographic references for interested readers on the subject.
1st paragraph in section Materials and methods (rows 141 – 142)
This sentence was removed.
2.1.1, 2.1.2, 2.1.3 and 2.1.4 (rows 196 – 240) - Information in these paragraphs is at the level of the first year of geoinformatics studies
We decided to keep those contents because it is useful as a reference for some readers not familiar with the mathematical form of geometric transformations. Besides, another reviewer found those sections convenient.
1st paragraph in section 3.3 Using a tile map service
We believe that Section 3.3 must be present in the paper. In 3.3 we describe de disk format which is a key characteristic of the tile map service (TMS) concept. Section 3.3 also helps in replying to other comments by this Reviewer #1 (the third one).
Other comments:
Title: “tile-based geospatial system” is very tricky term. Did you mean "raster GIS" or "tile-based GIS"?
We do not agree that ‘tile-based geospatial system’ is tricky terminology. It is a well stablished term in scientific or technical literature (for instance reference [18]). This kind of geospatial datasets are different from both raster GIS or tile-based GIS.
‘Raster GIS’ is a general term that does not capture some key characteristics of tile-based systems. Tile-based systems require splitting the whole dataset in multiple sub-images and levels of different resolution and detail in a hierarchical file structure. This information is given in Section 3.3 in the original manuscript.
We think that the term ‘tile-based GIS’ does not fit well to tile-based geospatial systems either. GIS has a very precise meaning in the geospatial community with well-defined characteristics regarding to spatial analysis functionalities that are nor present in tile-based systems. Tile-based systems are more of a map data repository kind, where analysis functionality must be added by specialised applications.
Title: This paper is not about "integration of the first 1:500 urban map", but about "integration of one map sheet"
We agree with the reviewer and think that changing the title can be convenient. We pose the following tile:
“A practical procedure to integrate the first 1:500 urban map of València into a tile-based geospatial system”
row 53: " ... 421 sheets of the project ..."; rows 478 - 479: " ... in 421 maps ..." Uniform terminology should be used. I think the correct term is a "map sheet(s)".
We made some edits to get uniform terminology, using always "map sheet(s)".
Material and methods: INSPIRE does not contains data quality rules, because it is out of the scope of this directive. On the contrary, national mapping agencies of different countries (also from Europe) use such standards.
We agree that geometric quality is out of the scope of INSPIRE. Since we do not know other quality standards in Europe, we used the one from the ASPRS which is well-known.
In our country (Spain) there is not any quality standards for large scale maps (1:500 and similar). There are specifications for scales 1:5000 and 1:10000 by the “Comisión de Normas Cartográficas” a branch of the Spanish National Mapping Council. The specification recommends appending information on data quality according to the Spanish Metadata Core, which describes a minimum set of elements describing the geographic information. That regulation does not apply to 1:500 maps.
3.2.1 Data points - Is this chapter about ground control points? The correct terms should be used.
Tables and their titles should be formatted better - Table 1 and Table 5.
Minor changes have been made to certain details of the tables to be formatted better, we have used the editorial template to make the tables.
What is the source of the vector layer of building footprints in Fig. 7?
The source od the vector data is the spatial data infrastructure of the Instituto Geográfico Nacional, the Spanish National Mapping Agency.
I personally disagree with the last sentence in the penultimate paragraph. The created application is a proof-of-concept, but the real potential or feasibility of this system is for me very low.
As stated by the reviewer, we think this a personal view of her/himself. The small viewer provided in the paper is a proof of concept which by definition is a small sample of a bigger system. Our proposal has solid ground, both theoretical and practical, and the only limiting factor is to get human and funding resources.
Regarding the feasibility of the system, conversations with regular users of the data (city council staff) show the high potential of the system, and actually they regularly use paper copies to solve even legal matters in real situations.
TMS (e.g. row 498) or Tms (row 516)?
It must be TMS.
Reviewer 2 Report
The objective of this manuscript is to establish an approach for building modern geospatial database from paper printed old maps that had been prepared with high accuracy techniques of the time.
The paper introduces three major steps of digitalization, georeferencing and rasterification, that each have been discussed in great detail.
The manuscript is written in a simple language that can be well understood by people who are not professionals in this field.
The title has been chosen properly to describe the content of the paper
The abstract is well written, and included all requirements of a journal paper abstract.
The study approach is clear and is well discussed in the manuscript.
Figures are of good quality.
Most elements in the manuscript have logically related the objective of this research.
The discussions are sound and justified and follow the presented results
NOTE: The reason for choosing "reconsider after major revision" is communicated to the editor.
Author Response
Dear Reviewer 2:
We really appreciate your comments.
Reviewer 3 Report
This is a well written paper. I have just two comments:
why the transformation from the 1929 geodetic coordinate system to ETRS89 is not
performed in an analytic way and a two dimensional coordinate transformation is applied?
2. the paper findings would be more stable if results were based on the transformation of more that one map e.g. 54II
Author Response
Dear Reviewer 3,
Thanks very much for the careful readings of the first version of the manuscript. Please, find below our replies to this review.
This is a well written paper. I have just two comments:
why the transformation from the 1929 geodetic coordinate system to ETRS89 is not
performed in an analytic way and a two dimensional coordinate transformation is applied?
This is an interesting question that was answered in a previous study (see reference [10] of the manuscript). The coordinate system of the 1929 files was unknown, but we found out that it was a Tissot projection, which was very common in the first decades of the 20th century. However, the coordinate system of the printed maps was not exactly the Tissot system we found in the project files, and we proceeded as if it were a local Cartesian 2D system.
That being said, we used 2D transformations because the original paper sheets were fundamentally bidimensional (the only 3D information items are some contour lines in small areas of the maps). However, since we collected GNSS data in the previous study, we were able to compute a 3D local vertical Cartesian (LVC) transformation with the objective of comparing the horizontal projection of the LVC with the 1929 coordinate system. We used another target coordinate system, the UTM projected onto the ETRS89 reference system, for obvious reasons of practicality and common usage in large map urban maps, specifically in Spain.
In [10], there is an interesting comparison between the 1929-LVC and 1929- UTM/ETRS89 transformations. The reviewers can get the paper from the editor if necessary, we have provided it him.
We think that our approach is very rigorous and can be implemented in practical applications properly.
We do not fully understand what the reviewer means by “analytic way”.
Besides, the area of the whole map is 13.4 (E-W) km x 7.7 (N-S) and is located in a very flat area, so that the authors of the map considered a local, 2D reference system. Is should be said in favour of the authors of the map that our results confirm the choice of a 2D coordinate system.
the paper findings would be more stable if results were based on the transformation of more than one map e.g. 54II
The goal of our study was to set a procedure or methodology to convert early surveying maps printed on paper to a modern digital format. Probably, this point was not clearly explained in the first version of the manuscript. We seek to work it out in these replies.
The fact that we present one only sheet is not relevant for the purpose of the study. The difference between processing one sheet or multiple sheets is a ‘production’ issue and not a ‘research’ issue. Therefore, the fundamental finding of the paper is to find the best approach in terms of geometric transformations, not to produce a complete map for public use. We were very accurate to test several geometric transformations and select the one that worked the best.
The results of that exploratory work allowed us to establish a method to keep the quality of the original in the final digital map, which is known to be 10-15 cm according to previous research (reference [10]).
Currently, we are in conversations with the Ajuntament de València (the City Council Authorities) on behalf of our department at UPV. The objective of those conversations is to sign an agreement to provide means (staff and funding) to conduct all that production work and publish a completely operating mapping system. The link provided in the manuscript was just a quick resource for reviewers to get the grasp of the system.
We added new contents in lines 66-69, 88-90 and 96-99 of the Introduction section to improve the explanation and understanding.
Round 2
Reviewer 1 Report
I find the modifications and/or responses to most of my comments adequate. I have only following comments:
If I look at several authors' reactions regarding the number of processed maps sheets, I recommend modifying lightly the abstract, introduction and conclusions. The paper describes mostly a pilot study (processing of one map sheet etc.). So this fact should be mentioned in abstract (row 19) as well as in introduction (8th paragraph) and conclusions. In conclusion there can be also described how to process all map sheets in future.
Title: “tile-based geospatial system” is very tricky term. Did you mean "raster GIS" or "tile-based GIS"?
We do not agree that ‘tile-based geospatial system’ is tricky terminology. It is a well stablished term in scientific or technical literature (for instance reference [18]).
Title of reference [18] is “Tile-Based Geospatial Information Systems …”, why you omit the word "information" in your title? I personally consider the term “Geospatial Information System” appropriate.
I think, that some references in the list are not formatted correctly [22, 31], specifically names of the authors.
Author Response
Dear Reviewer
Thanks very much for the careful readings of the manuscript. The authors are sincerely thankful for your detailed comments and suggestions that helped us to considerably improve the manuscript.
Please, find below our replies to this review.
I find the modifications and/or responses to most of my comments adequate. I have only following comments:
If I look at several authors' reactions regarding the number of processed maps sheets, I recommend modifying lightly the abstract, introduction and conclusions. The paper describes mostly a pilot study (processing of one map sheet etc.). So this fact should be mentioned in abstract (row 19) as well as in introduction (8th paragraph) and conclusions. In conclusion there can be also described how to process all map sheets in future.
This suggestion has been include in the new versión of the paper. We added new contents in row 19, introduction and conclusions to improve the explanation and understanding.
Title: “tile-based geospatial system” is very tricky term. Did you mean "raster GIS" or "tile-based GIS"?
We do not agree that ‘tile-based geospatial system’ is tricky terminology. It is a well stablished term in scientific or technical literature (for instance reference [18]).
Title of reference [18] is “Tile-Based Geospatial Information Systems …”, why you omit the word "information" in your title? I personally consider the term “Geospatial Information System” appropriate.
Title has been modified, which now include the word “Information”.
I think, that some references in the list are not formatted correctly [22, 31], specifically names of the authors.
References have been modified.
Reviewer 2 Report
The manuscript as-is has enough quality to be considered for publication.
Author Response
Dear Reviewer
Thanks very much for the careful readings of the manuscript. The authors are sincerely thankful for your detailed comments and suggestions that helped us to considerably improve the manuscript.